# Global Gene Expression of Post-Senescent Telomerase-Negative *ter1*Δ Strain of *Ustilago maydis*

**DOI:** 10.3390/jof9090896

**Published:** 2023-08-31

**Authors:** Juan Antonio Sanpedro-Luna, Leticia Vega-Alvarado, Candelario Vázquez-Cruz, Patricia Sánchez-Alonso

**Affiliations:** 1Posgrado en Microbiología, Instituto de Ciencias, Benemérita Universidad Autónoma de Puebla, Puebla 72570, Mexico; juan.sanpedro@alumno.buap.mx; 2Instituto de Ciencias Aplicadas y Tecnología, Universidad Nacional Autónoma de México, Ciudad de Mexico 04510, Mexico; leticia.vega@icat.unam.mx; 3Centro de Investigaciones en Ciencias Microbiológicas, Instituto de Ciencias, Benemérita Universidad Autónoma de Puebla, Puebla 72570, Mexico; candelario.vazquez@correo.buap.mx

**Keywords:** telomerase, *ter1* mutants, transcriptome analysis

## Abstract

We analyzed the global expression patterns of telomerase-negative mutants from haploid cells of *Ustilago maydis* to identify the gene network required for cell survival in the absence of telomerase. Mutations in either of the telomerase core subunits (*trt1* and *ter1*) of the dimorphic fungus *U. maydis* cause deficiencies in teliospore formation. We report the global transcriptome analysis of two *ter1*Δ survivor strains of *U. maydis*, revealing the deregulation of telomerase-deleted responses (TDR) genes, such as DNA-damage response, stress response, cell cycle, subtelomeric, and proximal telomere genes. Other differentially expressed genes (DEGs) found in the *ter1*Δ survivor strains were related to pathogenic lifestyle factors, plant–pathogen crosstalk, iron uptake, meiosis, and melanin synthesis. The two *ter1*Δ survivors were phenotypically comparable, yet DEGs were identified when comparing these strains. Our findings suggest that teliospore formation in *U. maydis* is controlled by key pathogenic lifestyle and meiosis genes.

## 1. Introduction

Telomerase is the ribonucleoprotein complex that solves the end replication problem by replenishing the telomere repeated motifs lost after each replication round. Telomerase does this using its unique and essential reverse-transcriptase activity [1,2]. The core components of this enzymatic complex are a protein subunit called *te*lomerase *r*everse *t*ranscriptase (TERT), which is responsible for the catalytic activity, and a lncRNA called *te*lomerase *R*NA (TER) or *te*lomerase *R*NA *c*omponent (TERC) that serves as a scaffold for assembly of the holoenzyme and carries the RNA template for the synthesis of the telomere repeated motif [3]. Telomerase is the predominant pathway for telomere lengthening in most eukaryotic cells [1]; in the absence of the enzyme, the telomere shortens as replication rounds increase. In somatic cells of higher eukaryotes, the *TERT* gene is downregulated as the cellular differentiation program advances [4]. However, different quantities of telomerase activity have been detected in the embryo-growing tissues and remain present in metazoans’ highly proliferative cell lineages, such as in germ line tissues, stem cells, multipotential cells, and, unfortunately, also in cancer cells [5,6].

In *Saccharomyces cerevisiae*, telomerase activity is expressed at the late S-phase during an undetermined number of replication rounds [7]. This is because the yeast TLC1 RNA subunit is expressed in an S-phase-dependent fashion, whereas the catalytic subunit Est2 depends on TLC1 abundance [8,9]. When the telomerase activity is abolished in the *tlc1* cells, the telomeres shorten and senescence sets in.

When a critical telomere-shortening threshold is reached, most of the cells stop growing and die; at any point after this, in the post-senescent culture, survivor cells may arise [10,11]. In these survivor cells, the DNA recombination mechanisms enable the enlargement of the chromosome termini via two routes known as alternative lengthening of telomere (ALT). Type I survivors are characterized by the over-amplification of Y’ subtelomeric elements interposed by small fragments of telomeric repeats, while type II survivors present long heterogeneous tracts of telomeric repeats [12,13]. Multiple research groups have investigated the mechanisms involved in triggering each survivor type [14,15]. The type I survivors rely on Rad52 activity, in addition to that of Rad51, Rad54, Rad55, Rad57, Rif1, Rif2, and nine other non-telomeric proteins. Lack of any of them increases the type II /type I survival rate. Type II survivors depend on Rad52 and Sgs1 helicase, as well as on Mre11, Rad50, Xrs2, Rad59, Tel1, Mdt1, Def1, Clb2, and Sua5 and 22 other non-telomeric proteins [15]. Epigenetic mechanisms seem to also be involved in controlling the expression of these genes [14].

Transcriptome analyses of *tlc1* mutants of *S. cerevisiae* along the cell progression (pre-senescent, senescent, and post-senescent cell stages) have been achieved to identify the network of genes involved in cell surviving in the absence of telomerase [7,16]. In those studies, several genes, named telomerase-deletion response (TDR) genes, were found to be deregulated, with some of them displaying a log2 fold change (logFC) in transcriptional expression ≥ 2 [7]. Among those TDRs genes, many are involved in responses to heat shock, MMS, and environmental stresses. Moreover, the global transcriptome of telomerase-negative mutants of diverse species is characterized by the expression of a related subset of genes involved in several abiotic stress responses, earlier named the core environmental stress response (CESR) genes when identified in *Schizosaccharomyces pombe* [17]. Additionally, in other lower eukaryotic species, the CESRs are also deregulated when telomerase activity is depleted [18,19], but still, a small subgroup of upregulated genes, which are rarely shared in other genotoxic responses, was an unique hallmark of telomerase deprivation in *S. cerevisiae* and have been named telomerase-deleted signature (TDS) genes [16].

TDR includes genes from carbohydrate metabolism, protein synthesis, folding, membrane structure, membrane trafficking, and genes harbored in subtelomeric sequences. The wider number and deepest response of TDR genes are attained when chromosomes reach the shortest telomere length; then, most of the genes return to a middle stage of deregulation in the survivor phase of the cells [16]; more recently, it has been reported that the TDR also comprises the deregulation of some ncRNA genes [20].

Multiple research groups have successfully addressed the efforts to elucidate the details of telomere metabolism in the basidiomycetous model organism *U. maydis*, which has reported the telomere structure and drafted its organization [21,22,23,24]. The two core subunits of the telomerase complex have been identified and partially characterized [22,25]; in addition, several hypothetical genes of the shelterin complex subunits have been identified, and some genes of the recombination machinery involved in chromosomal end maintenance have been characterized [23,24,26].

The TER subunit of the *U. maydis* telomerase is located on chr 8 as an overlapping sense lncRNA spanning the last exon of *UMAG_03168* and the proximal intergenic sequence *ter1* of 1626 bp (GenBank TPA: BK059259). The length of the gene has been calculated as at least 1280 bp; it transcribes more than one isoform, but the characterization of a full set of transcript products remains to be completed [25].

The shared telomere-maintenance pathways between *U. maydis* and *S. cerevisiae* prompted us to study the transcriptome of two *ter1*-disrupted *U. maydis* survivor strains. These strains lack the essential template domain. We set out to describe the variance in gene expression between *ter1*Δ and WT strains. This objective was chosen for two reasons: (1) TER (telomerase RNA) is the limiting subunit dictating telomerase functionality in yeast; and considering (2) in *U. maydis*, additional genetic constituents facilitate and regulate the transition from a yeast-like to a mycelial form.

Here, we report the global transcriptome analysis of two *ter1*Δ strains of *U. maydis*. Many of the identified TDRs are shared with other TER-disrupted fungi, but we also identify deregulated genes involved in meiosis, pathogenic development, and pathogen–plant crosstalk. Although a similar pattern of differentially expressed genes (DEGs) was found in both *ter1*Δ strains, they were not identical. In both strains, some genes involved in the pathogenic-mycelium lifestyle were deregulated in sporidia, suggesting an early interference with their ability to cross-fertilize with compatible strains and complete the life cycle.

## 2. Materials and Methods

### 2.1. Strains and Culture Medium

The *U. maydis* 518 (*a2b2*) strain was previously donated by WK. Holloman (Cornell Weill Medical Center, New York, NY, USA), the telomerase-negative derivative strains ter1-02 and ter1-24 (*a2b2*, *ter1::hph*) were constructed and characterized as previously described in [25]. The parental strain was used as a control. All the strains were grown in YEPS medium [1% yeast extract (Difco™ Thermo Fisher Scientific, Pittsburg, PA, USA], 1% tryptone peptone (Difco™ Thermo Fisher Scientific, Pittsburg, PA, USA), 2% sucrose (Merck Sigma-Aldrich, Burlington, MA, USA)] at 28 °C, supplemented with hygromycin (Merck Sigma-Aldrich, Burlington, MA, USA; 150 μg/mL) when necessary.

### 2.2. RNA Isolation and Transcriptome Sequencing

RNA extraction was performed as previously described [25]. For the elaboration and sequencing of cDNA libraries, the services of the Laboratorios de Servicios Genómicos (LabSerGen) de la Unidad de Genómica Avanzada, Laboratorios Nacionales de Genómica para la Biodiversidad (UGA-LANGEBIO) of the Centro de Investigación y de Estudios Avanzados del IPN (CINVESTAV) were used. The cDNA libraries were prepared using the TruSeq RNA Sample Preparation v2 LT kit (Illumina Inc. San Diego, CA, USA), and sequencing was performed on the Illumina NextSeq 500 platform, generating reads of 150 bp.

### 2.3. Data Analysis

The quality of the RNA-Seq reads was evaluated with the FastQC v0.11.7 software. Trimming of adapters and remotion of poor-quality reads was performed with Trimmomatic v0.36 [27], keeping the read-pairs with values greater than 25 nt in length and quality greater than 30 for the analysis. As part of the Trinity v2.11.0 pipeline [28], reads were aligned to the coding sequences (CDS) of the genome of the 521 strain, downloaded from the Ensembl database (https://ftp.ensemblgenomes.ebi.ac.uk/pub/fungi/release-56/fasta/ustilago_maydis/cds/Ustilago_maydis.Umaydis521_2.0.cds.all.fa accessed on 30 June 2021) using Bowtie2 v2.4.2 [29]. Next, the quantification of transcript abundances was done with the RSEM v1.3.3 program [30]. For the differential expression analysis, the count matrices generated in the previous step were uploaded to the IDEAMEX online platform (http://www.uusmb.unam.mx/ideamex/ accessed on 30 June 2021; [31] to use DESeq2 [32], NOISeq [33], Limma-Voom [34], and EdgeR [35], the cutoff values used to consider the genes as differentially expressed were logFC ≥ 2 and FDR ≤ 0.01. Raw sequencing reads, and expression data were submitted to the Gene Expression Omnibus of NCBI (accession number: GSE225422).

### 2.4. Annotation and Classification of Differentially Expressed Transcripts

The annotation of the resulting transcripts and products was carried out using the Trinotate v3.2.2 pipeline [36]; the sequences were analyzed by use of the BLAST+ 2.12.0 software packages [37] and the databases UniProt, Gene Ontology (GO), Kyoto Encyclopedia of Genes and Genomes (KEGG), and EggNog, applying the cutoff e-value of 10^−5^. Next, protein domains were sought in the outcomes employing the hmmer v3.3.1 software of Trinotate suite and the database Pfam-A. Likewise, SignalIP v4.1 and TMHMM v2.0c programs were used to seek the cleavage sites of signal peptides and to predict transmembrane helix [38,39,40]. DEGs were selected, and heatmaps were plotted using the R language. Finally, using the OmicsBox v1.4.12 pipeline (Bioinformatics Made Easy, BioBam Bioinformatics, Valencia, España 3 March 2019, https://www.biobam.com/omicsbox accessed on 30 June 2021), the GO identifiers of DEGs were filtered by taxonomy (fungi) and classified by functional annotation.

## 3. Results

### 3.1. Sequencing and Quality Control of RNA-Seq Libraries

The global expression profile of TER-deleted mutants at the surviving stage was analyzed using RNA-seq libraries of the parental strain 518 and its telomerase-negative derivatives ter1-02 and ter1-24. The assays were elaborated in independent triplicate repeats and sequenced using the Illumina NextSeq 500 platform (San Diego, CA, USA), yielding a total of 126,587,419 paired reads from nine libraries. After raw read processing, a total of 110,227,520 paired reads were preserved (Appendix A). The alignment percentages of the processed reads against the reference sequences with Bowtie 2 are shown in Appendix A.

### 3.2. Changes in the Expression Profile of Telomerase-RNA-Deletion Response (TDR) Genes

To identify the DEGs between the parental strain and *ter1*Δ mutants, the reads mapping to the reference transcripts were used for the analysis of differential expression on the IDEAMEX platform, setting up cutoff points of: logFC ≥ 2 and FDR ≤ 0.01, and only the genes appearing at the intersection of DESeq2, NOISeq, Limma-Voom, and EdgeR were considered differentially expressed. Likewise, the changes in the expression profile between ter1-02 vs. ter1-24 mutants were compared to find the differences exhibited by both mutants [25].

Concurrently with the differential expression analysis, the input data were analyzed to detect sample variations. In Figure 1, replica sets of each strain analyzed are shown in the multi-dimensional scaling (MDS) graph; as they were grouped in separate clusters, variations in the expression patterns were confirmed for each sample. From 6783 protein-coding genes analyzed, 241 (3.55%) were detected as overexpressed DEGs when compared WT and ter1-02 transcriptomes, whereas 276 DEGs were identified after comparing WT and the ter1-24 transcriptomes. Of these, 246 genes (3.62%) were overexpressed, and 30 (0.44%) were repressed in the mutant strain. To complete the analysis, a comparison of ter1-24 and ter1-02 transcriptomes led us to discover 211 (3.11%) DEGs in the ter1-24 mutant relative to ter1-02, of which 70 genes (1.03%) were overexpressed and 141 (2.07%) were repressed (Figure 2).

Once identified, the DEGs were retrieved and examined to identify the sets of unique and shared genes from each analysis. In *U. maydis*, deletion of TER causes the deregulation of a core of 90 genes shared by the two TER-mutants, hence named core genes; this represents 1.32% of the total coding genes. Eight of those genes showed clear differences in transcriptional expression among the mutants. When the search for unique DEGs was carried out for each pair of transcriptomes analyzed, 57 genes were identified for ter1-02, and 124 genes were identified for ter1-24. When the differential analysis was conducted against wild-type, 47 unique DEGs were identified when comparing ter1-24 with ter1-02 (Figure 3A).

### 3.3. Annotation and Functional Assignment

To examine how DEGs contribute to the survivor phenotype in *U. maydis*, the core genes and DEGs recovered from the former analysis were forwarded for functional annotation and GO analysis using the Trinotate pipeline and OmicsBox, respectively. For the DEGs from ter1-24 vs. the WT 518 strain, it was found that ≈36% of the genes do not share similarity with any of the sequences deposited in the protein databases accessed and therefore were considered to not have an assigned function. Regarding the DEGs from a comparison of ter1-02 vs. 518 transcriptomes, the percentage of genes without assigned function increased to ≈41% (Figure 3B). From the annotation of the total set of coding genes, 26% corresponded to genes without any function assigned. Thus, the role of more than a quarter of the coding genes remains to be elucidated; even more interesting is the finding that these genes only represent a small percentage of the global transcriptome of *U. maydis* previously calculated by [41], and by our group [25].

To determine which of the cellular functions were affected by TER deletion, DEGs were grouped into the three categories of the GO analysis: biological process (BP), molecular function (MF), and cellular component (CC). Figure 4 shows the ten most representative classes of each category of the GO for the DEGs identified in both *ter1*-disrupted mutants. In the BP category, the highest proportion of overexpressed DEGs matches with genes involved in cellular processes (82 DEGs in ter1-02, 100 DEGs in ter1-24) and metabolic processes (87 DEGs in ter1-02, 97 DEGs in ter1-24) followed by genes involved in localization and response to stimuli (25 and 24 DEGs in ter1-02, and 28 and 30 DEGs in ter1-24, respectively). In the case of downregulated DEGs from ter1-24, GO analysis showed 23 downregulated DEGs, which are involved in cellular processes, 15 DEGs in metabolic processes, and ten DEGs in cellular localization as the most representative categories; remarkably, there also were downregulated genes from multi-organism processes and stimuli response (five and three DEGs, respectively).

Regarding the MF category of the GO analysis, the main enrichment of upregulated genes in ter1-02 occurred in the categories of catalytic activity (95 DEGs), binding capability (69 DEGs), and transporter activity (22 DEGs); likewise, in ter1-24, overexpressed genes also were from catalytic activity (99 DEGs), binding capability (81 DEGs), and transporter activity (22 DEGs). Downregulated DEGs in the same MF categories were found in the ter1-24 strain with catalytic activity (15 DEGs), binding capability (13 DEGs), and transporter activity (10 DEGs). Of the overexpressed genes in ter1-02, it was interesting to observe that 41.05% (39) of those assigned in the status of “catalytic activity” have hydrolase activity, and 38.84% (35 genes) possess oxidoreductase activity; a similar enrichment of these two activities was found in the ter1-24 DEGs reaching 31.31% (31 genes with hydrolase activity) and 38.83% (38 genes with oxidoreductase activity).

In the CC category of upregulated genes, the most enriched DEGs of ter1-02 mutant were intracellular anatomical structure (66 DEGs), organelle (49 DEGs), and cytoplasm (47 DEGs). In the ter1-24 strain, the most enriched DEGs also were intracellular anatomical structure (81 DEGs), organelle (66 DEGs), and cytoplasm (64 DEGs). Likewise, in ter1-24, there were also downregulated DEGs in the same categories; the intracellular anatomical structure was enriched with a count of (13 DEGs), organelle (ten DEGs), and cytoplasm (nine DEGs). Finally, from the DEG analysis for the BP category in both mutants, the most abundant deregulated genes included those from the cellular process, metabolic process, localization, and response to stimuli. Agreeing with the above results, those DEGs were assigned to the catalytic activity group, genes that regulate molecular function, binding and transporter activity in the MF category. Interestingly, genes that regulate molecular function were among the most enriched DEGs (26.54%) obtained from the analysis of both mutant strains. For the CC category, the main enriched DEGs were assigned to the intracellular anatomical structure group, and the number of DEGs assigned to those located in the membrane group increased, as well as those assigned to organelles and intrinsic components of membranes (Figure 5).

### 3.4. Chromatin Structure

Within the DEGs in ter1-24, we found that *UMAG_02709* was repressed (logFC = −2.26), a locus that codes for the histone replication-coupled H3.2; repression of other core histone complex components was also noted (H2B (*UMAG_01505*), H2A (*UMAG_01504*), and H4 (*UMAG_02710*)) and were also found with logFC values of −1.98, −1.93, and −1.74, respectively, in addition to the putative H1 (*UMAG_10447*) with logFC value −1.83. However, although repression of the same genes was observed in the ter1-02 mutant, such changes were not significant at the cutoff point used, ranging between −0.93 (*UMAG_01504*) and −0.62 (*UMAG_10447*). Additionally, in both mutants, there were no significant changes in the expression of H2A.Z (*UMAG_00469*) and histone replication-independent H3.1 (*UMAG_03916*). Regarding other components involved in chromatin remodeling, in ter1-24, the changes were subtle, below the threshold in the genes *UMAG_02567* (logFC = −1.51) and *UMAG_06201* (logFC = −1.19) encoding for the catalytic subunit of histone acetyltransferase type B and RuvB-like helicase1 respectively. However, they need to be further explored.

### 3.5. Subtelomeric Genes

*U. maydis* has multiple predicted telomere-linked helicase (TLH) isoforms. These include RecQ-like sequences UT5 (AF030886.2) and UT6 (AF030887.1) [42]. UT5 and UT6 have been categorized as *UTASa* and *UTASb*, respectively. *UTASa* is adjacent to telomere repeats and possess the same helicase domain as the RecQ helicases subfamily. *UTASb* is near telomere repeats or 5’ upstream of *UTASa. UTASb* may be a truncated segment generated from the homing endonuclease gene HEG [43]. *UTASb* consists of a highly conserved “core” sequence, an OrsD domain, or both [42]. UT5 exhibits homology with *UMAG_12032* and *UMAG_12076*, while portions of UT6 are conserved in *UMAG_06476*, *UMAG_06474*, *UMAG_11065*, *UMAG_04094*, and *UMAG_04308*. *UMAG_12032* was upregulated in both mutant strains ter1-02 (logFC = 3.16) and ter1-24 (logFC = 5.20) (Table 1). The other putative proteins displayed heightened expression solely in ter1-02. *UMAG_12032*, situated at one end of chr 20, encodes an incomplete TLH1-like ATP-dependent DNA helicase. *UMAG_12032* may represent a novel secondary group of DEXH helicases that share homology with TLH1 [44]. The disruption of TLHs in telomerase-negative strains is frequently associated with gene de-repression due to its proximity to the shortened telomere [45] (Appendix A).

Transcriptional expression of genes arranged on the 20 kb adjacent to each of the 46 chromosome termini of *U. maydis* was analyzed to assess the effect of *ter1* disruption on them. These regions contain 146 genes. After analysis of DEGs in ter1-02, it was found that 30.13% of telomere-proximal genes (18.25% of the total DEGs) were differentially expressed, while in ter1-24, 10.27% of telomere-proximal genes were among DEGs (5.43% of DEGs), and from these only *UMAG_06259* (peptidyl-prolyl cis-trans isomerase D) was repressed. Seven genes were overexpressed uniquely in ter1-02 (*UMAG_05621*, *UMAG_11183*, *UMAG_06351*, *UMAG_11107*, *UMAG_04305*, *UMAG_10981*, *UMAG_04095*) with logFC greater than two, without being classified as differentially expressed in ter1-24 at the same cutoff point used. From the sets of DEGs located at the chromosomal ends, four belong to the core genes of the *ter1*Δ strains; among those are *UMAG_12032* and *UMAG_04695* that encodes a choline transporter and *UMAG_06146* and *UMAG_12031* that code for hypothetical proteins.

In addition to the DEGs from subtelomeric regions (Appendix A), we found other DEGs sheltered in three of the four unmapped scaffolds (accession NW_011929455.1 to NW_011929458.1). Set on NW_011929455.1, the *UMAG_06490*, which encodes a predicted inorganic phosphate transporter, was differentially expressed in ter1-24 (logFC = −2.21); in NW_011929456.1, the *UMAG_06503*, which encodes a DEG similar to the uncharacterized protein gene C1198.03c, had a logFC = 2.88; finally, placed on NW_011929457.1, the loci *UMAG_06504* (encoding a protein of unknown function) and *UMAG_12076* (encoding a protein related to ATP-dependent DNA Q5/DEAD/DEAH box helicase) were upregulated in ter1-02. The *UMAG_06504* fitted in the core of DEGs, and *UMAG_12076*, another RecQ-like helicase, was upregulated in ter1-02 with logFC = 4.93 but showed no change in ter1-24 (logFC = 0.22).

### 3.6. DEGs Related to Stress and DNA-Damage Response (DDR)

DEGs involved in stress and DNA-damage responses are listed in Table 2 and Table 3. Within the stress-response category, DEGs were mainly related to the oxidative stress response and cellular detoxification, except *UMAG_03122* (encoding a β-1,3-glucan-binding related protein), located within the first 20 kb of chromosomal ends. Most of these genes could be analogous to stress-induced responses and cellular senescence of *S. cerevisiae* [7]. The group also included the predicted genes implied in apoptosis, i.e., *UMAG_01937*, *UMAG_02224,* and *UMAG_03728*. In addition, *UMAG_04553* (predicted growth hormone-inducible transmembrane protein; logFC = 2.02) was differentially expressed in ter1-24 because of a slight downregulation in ter1-02, whereas *UMAG_11483* (predicted protein-L-isoaspartate (D-aspartate) O-methyltransferase) and *UMAG_01262* (predicted β-DNA-polymerase) were uniquely upregulated at logFC > 2 in ter1-24 but induced in ter1-02 at logFC = 1.15 and 1.83, respectively.

Among the DEGs in the DDR category (Table 3), the MutS4/Msh4 (*UMAG_12336*) and MutS5/Msh5 (*UMAG_12155*) homologs were annotated as members of the DEGs core genes (Figure 6); in yeast, those proteins are part of the ZMM group (Zip1-4, Msh4/Msh5, Mer3), which together with Mlh1 and Mlh3 promote crossover during meiotic recombination [47,48,49].

**Table 3 jof-09-00896-t003:** DNA-damage response genes affected by *ter1*Δ mutation. The DEGs obtained between the various comparisons are indicated in bold.

Gene Id	Description	WT 518 vs. ter1-02	WT 518 vs. ter1-24	ter1-02 vs. ter1-24
** *UMAG_12155* **	Msh5—MutS protein homolog 5 ^2^	**2.426**	**3.797**	1.398
** *UMAG_12336* **	Msh4—MutS protein homolog 4 ^2^	**3.674**	**2.454**	−1.205
** *UMAG_10845* **	Related to G/U mismatch-specific DNA glycosylase	1.183	**2.608**	1.444
** *UMAG_05917* **	Related to cryptochrome DASH	1.200	**2.370**	1.199
** *UMAG_01262* **	Related to DNA polymerase beta	1.834	**2.397**	0.585
** *UMAG_03290* **	Rad51—DNA repair protein RAD51 ^1^	**2.046**	0.765	−1.256
** *UMAG_11008* **	Mer3—ATP-dependent DNA helicase MER3 ^2^	**2.011**	**2.427**	0.432
** *UMAG_04165* **	Related to replication factor A protein 3	1.741	−0.387	**−2.098**
** *UMAG_01952* **	Related to UV-damage endonuclease	1.812	**2.683**	0.889
** *UMAG_00172* **	Related to meiotic recombination protein rec8	1.749	**2.187**	0.481

^1^ [50], ^2^ [51].

wAn increase in expression of Mer3 (*UMAG_11008*) homolog also occurred in both mutants (except for the NOISeq normalization value in ter1-02, logFC = 1.98; Appendix A); Mlh3 (*UMAG_03481*) and Mlh1 homolog (*UMAG_05208*) were upregulated more than two-fold in ter1-02. However, no such expression changes of *UMAG_05208* occur in ter1-24 (Appendix A). In this last strain, upregulation occurred to a poor extent in *UMAG_10845* (predicted to encode thymine-DNA glycosylase; logFC = 1.18), *UMAG_05917* (predicted to encode deoxyribodipyrimidine photolyase; logFC = 1.20), and *UMAG_01262* (predicted to encode β-DNA polymerase; logFC = 1.8), which did not reach significance. Further study is now needed.

Regarding the homologs of the yeast *RAD52* epistasis group in *U. maydis*, only *rad51* was differentially expressed (logFC ≥ 2) in the ter1-02 strain, whereas in ter1-24, overexpression of the same gene reaches only a logFC = 0.76. Also, we detected a slight upregulation of the genes encoding the RPA-heterotrimer: *RFA1* (*UMAG_05156*; logFC = 1.56), *RFA2* (*UMAG_02579*; logFC = 1.50) and *RFA3* (*UMAG_04165*; logFC = 1.74).

Homologs of the RAD3 epistasis group were differentially expressed in *U. maydis*. The putative *rad7* (*UMAG_00657*) showed an upregulation of nearly three times in the ter1-02 strain and of nearly two times in ter1-24 (Appendix A), whereas the *rad16* homolog (*UMAG_03263*) did not exhibit expression changes. Finally, in ter1-24, the repression of some genes encoding DDR constituents at values above the logFC threshold of −2 was considered in this report because the interaction of their protein products with those of each other suggests alterations in DNA replication and/or repair among repressed DEGS: PCNA (*UMAG_05403*, logFC = −1.18), DNA Polε catalytic subunit A (*UMAG_01008*, logFC = −1.56), and DNA ligase1 (*UMAG_11196*, logFC = −1.05). Regarding components of the heteropentameric RFC homologs in *U. maydis*, putative RFC5 (*UMAG_00920*) was downregulated marginally by logFC = −1.45; RFC4 (*UMAG_00729*) had a logFC = −1.05, and MCM10 (*UMAG_10135*) had a logFC = −1.25. On the other hand, the putative MCM2-7 complex exhibited logFC values between −0.62 and −1.01, except MCM5 (*UMAG_05064*) logFC = −0.10, which could also be considered an ambiguous value. Finally, the following genes encoding the homologous subunits of the DNA-directed RNA polymerase II complex were slightly downregulated: RPB11 (*UMAG_02324*; logFC = −1.83), RPABC1 (*UMAG_10512*; logFC = −1.25), RPABC2 (*UMAG_10433*; logFC = −1.01), and RPABC3 (*UMAG_04460*; logFC = −1.15) (Appendix A). Furthermore, their role in the telomerase-deletion response need to be deciphered using alternative strategies.

### 3.7. Genes Involved in Telomere Maintenance

#### 3.7.1. Complexes of Shelterin, CST, and MRX/MRN

Except for Tpp1, which showed near to 50% repression in the ter1-24 mutant, no significant changes were found in the expression of genes encoding the components of the shelterin complex (Trf1, *UMAG_02326*; Rap1, *UMAG_04676*; Tpp1, *UMAG_11538*; Pot1, *UMAG_05117*) [23,26], CST complex (Stn1, *UMAG_11687*; Ten1, *UMAG*_11842) [23], or the putative MRX/MRN complex (Mre11, *UMAG_04704*; Rad50, *UMAG_01085*) [26].

#### 3.7.2. Putative SM7-like Subunits

A highly conserved motif has been found at the 3′ end of the putative TER of all the members of Ustilaginales studied in [25], which share homology with the binding site for Sm7. This finding, which suggests the requirement of an SM7-like complex for telomerase biogenesis, prompted us to search the Ustilaginales’ genes encoding the protein homologs of B/B’, D1, D2, D3, E, F, and G and analyze their expression changes in the *ter1*Δ mutants of *U. maydis.* From those putative homologs of SM7 genes, SMD1 (*UMAG_10381*, logFC = −1.57), SMD2 (*UMAG_04781*, logFC = −1.26), and SMD3 (*UMAG_11043*, logFC = −1.20) were slightly downregulated; a marginal change was found in SMB (*UMAG_12244*, logFC = −0.75), but no expression changes were found in SME (*UMAG_10312*), SMF (*UMAG_12130*), and SMG (*UMAG_10805*) in the ter1-24 mutant. The same strain also showed a slight downregulation (i.e., logFC < 2.0) of components of the predicted Dyskerin complex: Dyskerin/CBF5 (*UMAG_00685*, logFC = −1.18), NPH2 (*UMAG_03340*, logFC = −1.61), GAR1 (*UMAG_04573*, logFC = −1.03); and NAF1 (*UMAG_03271*, logFC = −0.87); no expression changes were predicted for NOP10 (*UMAG_03354*).

#### 3.7.3. Telomere-Linked Helicases (TLH1-like)

As noted above, the participation of middle repeated elements *UTASa* and *UTASb* harboring ORFs encoding helicase-like sequences in telomere maintenance has been proposed by some authors [44,52]. In both the ter1-02 and ter1-24 mutant strains, the *UMAG_12032* locus was upregulated; this could be associated with a possible gene de-repression of subtelomeric sequences after telomere-repeats loss or a possible role of this RecQ-like helicase in the primary DNA metabolism in the absence of telomerase [52] (Appendix A). Interestingly, the expression of *UMAG_12032* in ter1-24 is sufficiently high compared to ter1-02 to be considered as a DEG (logFC = 2.06) and is the only helicase-related gene overexpressed in this strain in contrast to the seven genes observed in ter-02 (Table 2). The upsurge in transcriptional expression of such genes in *U. maydis* needs to be surveyed to determine if they play a role in the formation of survivors or if their activity could be dispensable.

### 3.8. Genes Involved in Cell Cycle Progression and Pathogenic Development

The dimorphic fungus *U. maydis* requires *in planta* development to complete its life cycle. To achieve this, it experiences a morphologic and metabolic transition from saprophytic sporidia (yeast-like) to pathogenic mycelium (hyphae); this lifestyle change requires the reprogramming of gene expression [46,53,54]. Genes engaged in controlling the cell cycle and the fungal growth within the plant tissues, as well as genes encoding effectors required to subdue the plant response, are critical to complete the life cycle of this obligated fungal pathogen. The loss of key genes involved in the development or maintenance of the pathogenic lifestyle could cause growth distress and impairment to complete the life cycle [46,55,56].

#### 3.8.1. The Cell Cycle Progression

In the *ter1*Δ mutants, there were no expression changes detected in the master regulator *cdk1*, nor in the *cdk5* involved in cell differentiation [57,58], yet Cdk5-related cyclin-partner *pcl12* [59] reached the DEG threshold only in ter1-24; the same gene had a logFC = 0.86 in ter1-02; *pcl12* and *rbf1* are controlled by the heterodimeric master regulator bE/bW; however, its expression pattern suggests an independent mechanism of control [60] (see Table 4).

**Table 4 jof-09-00896-t004:** The life cycle and pathogenic development genes are deregulated in response to the *ter1*Δ mutation. The DEGs obtained between the various comparisons are indicated in bold.

Gene Id	Description	WT 518 vs. ter1-02	WT 518 vs. ter1-24	ter1-02 vs. ter1-24
** *UMAG_10529* **	Pcl12—related to PHO85 cyclin-2 ^10^	0.860	**3.388**	**2.535**
** *UMAG_00628* **	Putative protein of unknown function ^8^	**5.109**	**—**	−1.462
** *UMAG_00876* **	Related to glucan 1,3-beta-glucosidase ^1^	0.900	−1.435	**−2.31**
** *UMAG_01130* **	Related to tyrosinase ustQ ^7^	−0.164	**2.628**	**2.801**
** *UMAG_01237* **	Putative protein of unknown function, cluster 2A ^1^	**5.519**	**—**	**−5.015**
** *UMAG_01238* **	Putative protein of unknown function, cluster 2A ^1^	**6.249**	**—**	**−5.815**
** *UMAG_01431* **	Fer6—multidrug resistance protein fer6 ^2^	−1.163	**−3.861**	**−2.674**
** *UMAG_01432* **	Fer5—acyltransferase fer5 ^2^	−0.767	**−3.711**	**−2.92**
** *UMAG_01433* **	Fer4—enoyl-CoA isomerase/hydratase fer4 ^2^	−0.613	**−6.571**	**−5.924**
** *UMAG_01690* **	Putative protein of unknown function ^7^	**3.949**	**—**	**−2.207**
** *UMAG_01695* **	Stp6—Putative protein of unknown function ^10^	**—**	**5.615**	0.842
** *UMAG_01788* **	Related to chitin deacetylase ^3^	1.022	**−2.465**	**−3.476**
** *UMAG_01829* **	Related to alpha-L-arabinofuranosidase A ^1^	**2.673**	1.529	−1.126
** *UMAG_01888* **	Probable serine carboxypeptidase, cluster 3A ^1^	0.532	**2.143**	1.633
** *UMAG_01945* **	Putative invertase ^3^	**2.373**	**4.817**	**2.465**
** *UMAG_02135* **	Effector family protein Eff1-5 ^6^	**3.740**	**2.215**	−1.498
** *UMAG_02136* **	Effector family protein Eff1-6 ^6^	**7.088**	**—**	−1.341
** *UMAG_02137* **	Effector family protein Eff1-7 ^6^	**5.394**	**4.716**	−0.649
** *UMAG_02138* **	Effector family protein Eff1-8 ^6^	**3.099**	**2.239**	−0.820
** *UMAG_02140* **	Effector family protein Eff1-10 ^6^	**2.426**	**2.692**	0.301
** *UMAG_02758* **	Putative protein of unknown function ^3^	−0.473	**−2.133**	−1.632
** *UMAG_03023* **	Related to ribonuclease T2-like 1-A ^3^	−0.653	**−4.881**	**−4.193**
** *UMAG_03382 ** **	Related to 3-phytase A ^3^	1.305	**2.814**	1.534
** *UMAG_03411* **	Xin1—endo-1,4-beta-xylanase ^1^	**5.562**	**9.175**	**3.65**
** *UMAG_03416* **	Putative protein of unknown function ^1^	**2.482**	**2.625**	0.160
** *UMAG_03749* **	Putative protein of unknown function, cluster 10A ^1^	1.944	**2.601**	0.700
** *UMAG_03750* **	Putative protein of unknown function, cluster 10A ^1^	**—**	**4.125**	**2.386**
** *UMAG_03751* **	Putative protein of unknown function, cluster 10A ^1^	**5.598**	**3.721**	−1.824
** *UMAG_04282* **	Related to 3-phytase A ^3^	1.336	**3.869**	**2.556**
** *UMAG_04309* **	Probable alpha-L-arabinofuranosidase ^1^	**4.469**	0.820	**−3.625**
** *UMAG_04364* **	Related to glucan 1,3-beta-glucosidase ^1^	**2.904**	**2.711**	−0.162
** *UMAG_04503* **	Probable alpha-galactosidase D ^1^	**4.467**	**—**	**−3.390**
** *UMAG_04816* **	Egl3—Related to endoglucanase 1 ^1, 5^	**5.717**	**6.415**	0.698
** *UMAG_05036* **	Related to probable glycosidase C21B10.07 ^3^	0.744	**3.080**	**2.360**
** *UMAG_05299* **	Putative protein of unknown function, cluster 19A ^1^	0.756	**2.789**	**2.067**
** *UMAG_05306* **	Putative protein of unknown function, cluster 19A ^1^	**4.330**	**—**	−1.603
** *UMAG_05308* **	Putative protein of unknown function, cluster 19A ^1^	**2.876**	**—**	**−2.949**
** *UMAG_05310* **	Putative protein of unknown function, cluster 19A ^1^	**7.574**	**—**	**−9.239**
** *UMAG_05314* **	Putative protein of unknown function, cluster 19A ^1^	**3.579**	**—**	**−3.060**
** *UMAG_05361* **	Lac1—laccase ^3, 4^	**3.437**	**2.576**	−0.836
** *UMAG_05439* **	Related to GlcNAc-binding protein A ^7^	**3.655**	−0.031	**−3.665**
** *UMAG_05495* **	Related to papain inhibitor ^7^	**4.355**	**5.372**	1.037
** *UMAG_05689* **	Related to Fe-regulated protein 8 ^2^	**2.425**	1.702	−0.700
** *UMAG_05861* **	Lac2—laccase-2 ^3, 4^	**5.316**	**5.355**	0.064
** *UMAG_06190* **	Related to chitinase A1 ^3^	**3.479**	**—**	−1.840
** *UMAG_06221* **	Putative protein of unknown function, cluster 22A ^1^	**2.243**	1.030	−1.183
** *UMAG_06222* **	Putative protein of unknown function, cluster 22A ^1^	**6.296**	**4.759**	−1.521
** *UMAG_06274* **	Related to hormone-sensitive lipase ^3^	1.259	**3.221**	1.985
** *UMAG_06332* **	Egl1—endoglucanase 1 ^1, 5^	**5.283**	**3.747**	−1.518
** *UMAG_10055* **	Related to glutathione hydrolase proenzyme ^3^	**3.490**	**—**	**−2.091**
** *UMAG_10557* **	Putative protein of unknown function, cluster 19A ^1^	**4.454**	**—**	**−3.768**
** *UMAG_11338* **	Fer8—Fe-regulated protein 8 ^2^	0.141	**−2.720**	**−2.833**
** *UMAG_11339* **	Fer7—siderophore transporter fer7 ^2^	−1.095	**−4.791**	**−3.672**
** *UMAG_12330* **	Putative protein of unknown function ^3^	1.550	**3.223**	1.701
** *UMAG_05528* **	Related to alkali-sensitive linkage protein 1 ^9^	**5.131**	**—**	**−4.945**
** *UMAG_02161* **	Related to meiotically upregulated gene 190 protein	**2.322**	**2.503**	0.205
** *UMAG_02517* **	Gpa2—guanine nucleotide-binding protein alpha-2 subunit	1.110	**2.444**	1.360
** *UMAG_02994* **	Related to sporulation-specific protein 5	−0.065	**2.505**	**2.591**
** *UMAG_03541* **	Related to meiotic expression upregulated protein 26	**4.233**	**—**	−0.999
** *UMAG_05467* **	Related to meiotic coiled-coil protein 2	**5.627**	**—**	**−3.148**
** *UMAG_11677* **	Related to serine/threonine-protein kinase cek1	0.837	**2.533**	1.720

^1^ [46]. ^2^ [61]. ^3^ [62]. ^4^ [63]. ^5^ [64]. ^6^ [65]. ^7^ [66]. ^8^ [67]. ^9^ [68]. ^10^ [59]. * Located in subtelomeric region.

The analysis revealed a significant downregulation of Rad21 (logFC = −1.21). This Rad21 downregulation in ter1-24 went near half in the wild-type cell expression. The genes responsible for encoding the postulated cohesin proteins, Smc1 and Smc3, also exhibited substantial deregulation, with their normal expression levels decreasing by nearly half at logFC = −1.15 and logFC = −0.94, respectively. This downregulation could be correlated with the changes in cell morphology and nuclear irregularities observed in *ter1*Δ mutants [25], possibly indicating alterations in chromatin structure. Modest upregulations were noticed in the checkpoint kinase *chk1* (logFC = 1.40) and *wee1* kinase (logFC = 1.18) in ter1-02 and ter1-24, respectively, although both were below the logFC threshold of 2.0. Notably, no alterations were detected in the predicted apical kinase *atr1* expression within the ter1-02 and ter1-24 mutants (logFC = 0.51 and 0.49). This finding weakens the likelihood of a link between the apical members of DDR and the changes described above [69,70]. Similar instances have been reported in fission yeast and other Rad21-deficient cell lines like mouse embryonic fibroblasts (MEFs) and lymphoblastoid cell lines (LCLs) before [71,72,73].

Since the third portion of the DEGs in mutant strains lacks an assigned function or a defined role, a database of genes involved in pathogenic development was constructed, which included fungal secreted proteins, cell-wall-degrading enzymes, plant component-degrading enzymes, and proteins involved in the organization of the fungal cell wall, to further learn about the influence of chromosome architecture and fungal development [46,62,66,68,74].

#### 3.8.2. The Pathogenic Development

About 10% of the DEGs were involved in pathogenic development in both mutants. Those genes were included in the group of core genes and included some encoding plant cell wall-degrading enzymes such as *egl1* and *egl3* endoglucanases, *xin1* endoxylanase, which is engaged in the fungal cell wall organization, and *lac1* and *lac2* involved in melanin synthesis [63,64].

Particular gene clusters, which are collectively controlled, were upregulated in both mutants, i.e., the genes encoding the Eff1 protein family (except for *eff1*) located on the chr 5; those genes whose products have a role in virulence were upregulated [46,65] (Table 4), possibly, the lack of telomerase could cause deregulation of these genes through the modification of the expression pattern of key controllers of pathogenicity. Remarkably, except for *fer3*, the downregulation of the *fer* gene cluster, located on the telomere-proximal right end of the chr 2, also occurred in ter1-24 strain of *U. maydis* (Figure 7A), with *fer9*, *fer10,* and *fer11* being the most telomere-proximal genes [61].

Both the *fer* cluster as well as the *sid1*, *sid2*, *fer1*, and *fer2* genes are negatively regulated by the Urbs1 factor [61,76], but neither *urbs1* nor its target genes outside the *fer* cluster were downregulated, suggesting chromatin relaxation in the subtelomeric region could facilitate the interaction of Urbs1 with other negative regulators or with the *fer*-cluster promoters.

Moreover, the *UMAG_01436* (related to acetyltransferase MAT1) and *UMAG_01438* (predicted Acyl-CoA-dependent acyltransferase gene) loci, which separate *fer9-11* from the remaining genes of the *fer* cluster, were upregulated in ter1-24; whereas strong upregulation (logFC = 7.35) of the *UMAG_11874* gene, encoding a hypothetical protein of unknown function, was detected in ter1-02. Moreover, *UMAG_11874* is the last gene at the end of chromosome 2 (Figure 7A).

Other DEGs-encoding proteins of unknown function located within clusters 2A, 10A, 19A, and 22A [46] were deregulated in one or both mutants (Table 3). Finally, the genes involved in the evasion of the immune response and release of plant nutrients, such as phytases (*UMAG_03382*, *UMAG_04282*), lipases (*UMAG_06274*), and proteases (UMAG_12330) [62], were identified as DEGs in ter1-24; meanwhile, those in ter1-02 showed slight increases (1< logFC < 2).

### 3.9. Differentially Expressed Transcription Factors

The DEGs-encoding transcriptional factors (TFs) are listed in Table 5. Among the set TFs-genes altered in the *ter1*Δ mutants, some were associated with pathogenic development, such as *hdp1*, which was differentially expressed in both the ter1-02 and ter1-24 mutant strains. The *hdp1* is required for filamentous growth and cell cycle arrest [77]; likewise, *hdp2* (*UMAG_04928*), which plays a role in appressorium formation [64], was upregulated (below the threshold of two) in both mutant strains (logFC = 1.61 in ter1-02 and 1.81 ter1-24). The *fox1* gene, involved in suppressing plant defenses [78], was highly overexpressed only in ter1-02 (logFC = 4.98). The three TFs (*hdp1*, *hdp2*, and *fox1*) are activated by the Rbf1 master regulator (*UMAG_03172*), which in turn is activated by the bE/bW heterodimer in a dependent manner and also participates in the activation of *biz1* and MAP kinase Kpp6, which are involved in appressorium formation [64,79].

Also, *kpp6* was identified as differentially expressed in ter1-24, as was *rbf1.* Those results in sporidia, which lack the functional bE/bW heterodimer because they are haploid cells, together with the absence of deregulation, or with a slight increase in ter1-24, in the expression of factors upstream of Pfr1 suggest that other unidentified components or possibly Ter1 could be involved in the regulation of *rbf1*; and hence, some of the target genes of Rbf1 (Figure 8) as occurs when the fungus behaves as a necrotrophic pathogen, and in ter1-24 [60]. Other DEGs which encode transcription factors that are shared by the *ter1*Δ mutants as core genes were: the homolog of Yap (*UMAG_3296*) bZIP protein from the AP-1 family, involved in response to oxidative stress and plant–host interaction (Table 5; [82,83]), and *unh1*, the TF required for formation and pigmentation of teliospores, and completion of meiosis [63]. Intriguingly, none of its five target genes (*UMAG_00983*, *UMAG_06485*, *UMAG_04827*, *UMAG_05664*, *UMAG_11505*) were deregulated in either mutant, suggesting the cooperation of other unidentified factors in the regulation of these genes.

Among the TFs upregulated solely in the ter1-24 mutant were: *UMAG_01456*, *UMAG_06308*, *UMAG_02835*, and *UMAG_04778*, which in natural circumstances are upregulated during pathogenic development [80,84,85], and *UMAG_01025*, which is progressively repressed during early infection and responds to unfolded proteins [80,81]. Whereas in the same TFs category, the DEGs identified only in the ter1-02 were the *UMAG_12304* and *UMAG_04101* that encode Mtf1, a transcription regulator of the gene cluster located at the end of chromosome 12 involved in melanin production [75,86]; consistently, its target genes (*pks5*, *pks4*, *vbs1*, *orf4*, *pks3*, *omt1*, *pmo1*, and *cyp4*) were also identified as DEGs, while *orf1*, *orf5*, *aox1,* and *deh1* do not exhibit changes in transcriptional expression, nor *mtf2* or its target *orf2* (Appendix A).

Although no information about the expression of *mtf1* in ter1-24 was registered, upregulation of *vbs1*, *orf4*, *pks3*, and *pmo1* supported our notion that a weak upregulation of *mtf1* occurred in this mutant, as it was hinted in the analysis of gene expression between both mutants (Table 5). Alternatively, telomere loss could have alleviated the silencing of the subtelomeric gene cluster in ter1-24 (Figure 7B). Another insight was the induction of *pks1*, *pks2*, and *lac1* expression observed in ter1-02, but solely of *lac1* in ter1-24 mutant; those three genes play principal roles in teliospores melanization [87]; paradoxically, expression of *ust1*, —the TF regulator of *pks1*, *pks2* and *lac1—* remained unchanged in both mutants.

## 4. Discussion

Here, we present a transcriptomic DEGs analysis of three strains: the wild-type *U. maydis* 521 strain and two survivor strains. In both survivor strains, the template domain of *ter1* was disrupted. In *U. maydis*, *ter1* encodes the TER subunit of telomerase. In these *ter1*Δ strains, the template domain was replaced by the selective marker gene *hph*, as detailed previously [25]. This replacement did not lead to the suppression of transcriptional activity in the flanking genes *UMAG_03168* and *UMAG_03169*. The *ter1* transcript is initiated within *UMAG_03168*, a putative gene (Emi1) situated 5′ upstream from the intergenic region housing the conserved domains of the *ter1* gene [24]. Nevertheless, no polar effects were evident in the mutants; the A (unspliced) and B (spliced) isoforms were upregulated (Figure 6).

The *UMAG_03169* gene is at the 3′ end of *ter1* and encodes a putative ornithine-oxo-acid transaminase. *UMAG_03169* was also upregulated in the mutants (logFC = 1.74 in ter1-02 and logFC = 3.0 in ter1-24). This upregulation in these two genes cannot be solely attributed to the transcriptional disruption caused by ter1. Instead, it may be linked to the Telomere Damage Response (TDR). This would be consistent with observations for several other genes.

The putative telomere maintenance genes [26,88] were unaffected in *U. maydis*. This finding is consistent with previous studies addressing other telomerase-negative mutants [16,18,52,89]. However, *rad51* was upregulated (logFC = 2). The *rad51* gene is involved in the ALT pathways in *S. cerevisiae* and in DNA repair through homologous recombination. Conversely, the gene encoding its associated protein Brh2 (*UMAG_03200;* [69]) showed no changes in expression.

The chromatin modification pattern of telomere-associated sequences (TAS) shifts as telomeres shorten. This shift influences the condensation status of both the TAS chromatin and neighboring genes. Reducing chromatin compaction could lead to a variegated gene expression pattern, impacting genes close to telomeres like *UMAG_05721*. This gene encodes the *srbA* homolog in *U. maydis*, termed *srb1*, which plays a role in hypoxia adaptation and is a potential co-regulator of tumor formation [80]. Interestingly, in the ter1-02 strain, *UMAG_05721* is repressed, whereas *srb1* is upregulated in ter1-24 (Table 4).

Other genes within the two *ter*Δ mutants exhibited divergent transcriptional behaviors. However, as these strains share the same mutation, induce similar infection symptoms, and cannot produce tumors *in planta*, it is likely that a set of “core genes” orchestrates control over these strains during the early stages of their biotrophic development [25]. Additionally, our analysis was extended to several genes situated within a 20-kb range adjacent to the TAS. Some of these genes were deregulated (Figure 7), while others situated at identical distances were unaffected. This variation could be attributed to the penetrance of the *ter1*Δ mutation or the telomere-position effect (TPE), as previously discussed by [90]. The TPE might involve the repression of sequences neighboring the telomere in a telomere-length dependent fashion; in the case of TPE over long distances (TPE-OLD), it could govern the physical interaction of remote genes (up to 10 Mb away), silencing their expression without affecting neighboring genes [91]. The phenomenon of TPE has been identified in higher eukaryotes and various lower eukaryotes [92], playing pivotal roles in processes such as antigenic variation in protozoa, adaptation to rapid environmental changes in fungi, and replication, repair, and recombination processes [92]. Similar mechanisms could conceivably be at play within *U. maydis*.

Due to the TPE, it was previously believed that the ends of chromosomes were transcriptionally silent. However, recent findings challenge this notion. In organisms like *S. pombe* and various eukaryotes, transcripts originating from subtelomeric sequences have been observed. Notable examples include TERRA, ARRET, and α-ARRET, as well as transcripts like ARIA, that initiate from the telomere itself [93]. In both *S. pombe* and humans, TERRA and ARIA transcripts, which incorporate telomere repeats, display a lesser degree of polyadenylation compared to ARRET and α-ARRET, representing antisense and sense transcripts from subtelomeric regions [94].

For the transcriptomes of the *ter1*Δ mutants studied here, the poly(A) fraction was extracted from total RNA. Notably, a strand-specific approach was not employed for sequencing, presenting transcripts as homologous to subtelomeric sequences. Previously, in *U. maydis*, sense and antisense transcripts derived from chromosome ends were amplified using RT-PCR. Although sequences analogous to TERRA, ARRET, and α-ARRET were investigated by our group, their characterization remains incomplete, including determination of polyadenylation extent and percentage. Given that *UTASa* primarily localizes adjacent to telomere repeats, it is conceivable that their transcripts may lack polyadenylation, akin to the case of TERRA. This absence of poly(A) tail in transcripts could impede their detection in polyadenylated transcriptomes, thereby accounting for their underrepresentation in the analysis. Consequently, this could explain the prevailing abundance of *UTASb* sequences.

*UMAG_12032* and *UMAG_12076* are two upregulated *UTASa* sequences containing RecQ-like motifs. These *UTASa* sequences lack adjacency to telomeres. This suggests the possibility of these sequences undergoing transcription in a manner akin to ARRET or α-ARRET sequences. Also, *UMAG_11065* experiences an approximately 100-fold increase in expression within the ter1-02 strain. Both this sequence and *UMAG_06474* share homologous traits with *UTASb* and are notably detached from telomere repeats.

Moreover, the significance of TERRA extends to its role in telomere lengthening in scenarios where telomerase is absent. This is attributed to TERRA’s involvement in the formation of telomeric R-loops, which represent RNA/DNA hybrids. These structures facilitate recombination-based alternative mechanisms (ALT) that amplify chromosome ends. This phenomenon, in turn, ensures the survival of cells even in the absence of functional telomerase [95].

Non-telomeric functions have also been attributed to TERRA, shelterin proteins, constituents of the CST complex, and TERT, collectively referred to as telomere-associated factors (TAFs). These proteins have been found to intricately modulate the transcriptional activity of critical genes dispersed along the length of the human genome [96]. Additionally, the core components of telomerase have demonstrated functions beyond their telomeric roles across diverse organisms and cell lineages, although the primary focus has been on TERT (as reviewed in [97,98]). However, insights into the non-telomeric roles of TER are steadily emerging.

Consequently, it has come to light that the TER component engages in various non-telomeric activities. For instance, it plays a pivotal role in responding to DNA damage [99], offering protection against oxidative stress in motor neuron cells [100], and potentially regulating apoptosis by promoting cell survival through their intra-TER gene *hTERP* [101,102]. Furthermore, TER also regulates gene expression and cell differentiation [103,104,105,106]. Notably, a previous investigation unveiled a correlation between the effectiveness of telomerase function and the formation of teliospores [22].

Within *ter1*Δ mutants, the deregulation of at least 12 pivotal developmental TFs has been identified (Table 5). While the influence of TAFs on TF regulation in lower eukaryotes cannot be definitively ruled out, the attenuation of TPE is plausible, especially for TFs neighboring telomere repeats. As discussed above, chromatin relaxation could facilitate interactions between the repressor Urbs1 [61] and its recognition sites on the *fer* genes, subsequently downregulating gene expression. Subtle opposing fluctuations in *urbs1* expression (logFC of -0.38 in ter1-02, and logFC of 0.31 in ter1-24) appear to exert divergent and substantial impacts on the *fer3* to *fer8* genes within the strain. This effect is particularly prominent in the ter1-24 strain (Figure 7A).

The polyketide synthase (PKS) gene cluster at the terminal end of chromosome 12 (Figure 7B) encompasses 16 genes responsible for the synthesis of orsellinic acid and a melanin-like compound with dark pigmentation [75]. The regulatory pathways governing PKS biosynthesis are controlled by *mtf1*, a transcription factor within the PKS cluster. *mtf1* shares homology with the human *MYB* isoform 5. In circumstances where telomeres shorten, resulting in diminished TPE, *mtf1* expression is activated, ultimately leading to the upregulation of the entire cluster (Table 4). Consequently, this biochemical pathway might contribute to the distinctive brown pigmentation observed in *ter1*Δ mutants. However, further investigation is needed. 

The *bW* gene is also influenced by the disruption of *ter1*. *bW* and *bE* play a pivotal role in governing dimorphism and the pathogenic transition in *U.* maydis. These genes produce proteins that form a heterodimeric TF named bE/bW; both partners are needed for functionality [107]. Encoded within the *b* locus, these key components are transcribed from divergent promoters. Intriguingly, in the ter1-02 strain, solo expression of bW2 was observed. This occurrence echoes previous reports wherein solo *bW* and other genes exclusive to the pathogenic lifestyle were de-repressed in sporidia lacking the histone deacetylase *hda1*Δ [108]. The deregulation of solo *bW* reinforces the notion of epigenetic factors influencing the development of the pathogenic lifestyle within haploid sporidia mutants. However, beyond epigenetic influence, the atypical transcription at the *b* locus raises the intriguing question of how its target genes become de-repressed. An unconventional possibility could be the formation of functional homodimers by bW under exceptional circumstances in *U. maydis*, analogous to the situation in the distantly related basidiomycetous fungus *Cystofilobasidium capitatum* [109].

One of the prime targets of the bE/bW complex is *rbf1*, a transcription factor that governs the filamentous and pathogenic transition in *U. maydis*. In the ter1-02 strain, *rbf1* experiences a slight upregulation. Whereas in ter1-24, *rbf1* demonstrates more pronounced upregulation (Table 5). Additionally, Rbf1 governs Hdp1 and Fox1 (*UMAG_01224* and *UMAG_01523*, respectively). Within both mutants, Hdp1 displays elevated expression levels. Removal of *hdp1* produces filament elongation and G2 cell cycle arrest in wild-type cells [77] akin to the hyper-elongated cells and reduced growth rate observed in the *ter1*Δ mutants. Conversely, Fox1 regulates genes responsible for suppressing plant defenses through secreted effectors, some of which are harbored in clusters 10 and 19, such as *UMAG_03751*, *UMAG_05308*, and *UMAG_05314*. These genes were upregulated in the *ter1*Δ mutants, but it has been reported that deletion of Fox1 causes their downregulation [78]. Other members of clusters 10 and 19 are represented among the DEGs in Table 5.

Figure 8 illustrates a schematic representation of the central pathways controlling pathogenic development. We propose roles for the prominent TFs in the transition of *U. maydis*’s lifestyle and their contributions to the phenotypic changes and DEGs uncovered in this study. Notably, other factors not explicitly depicted, such as *unh1* (*UMAG_02775*; [63]), were upregulated and may have contributed to the inhibition of teliospore formation and meiosis. Similarly, *UMAG_01025* and *UMAG_12304*, implicated in the unfolded protein response (UPR; Table 5), were upregulated. As were *UMAG_02835* and *UMAG_02775* –linked to conidiophore development and meiosis (Table 5)– and several others [63,81]. Nonetheless, conducting individual analyses for each TF may have limited value, considering that the overexpression of transcriptional factors has been linked to growth inhibition in *S. cerevisiae* [110], suggesting a similar outcome in the mutants.

Equally noteworthy, in contrast to the *trt1*Δ mutants, *ter1*Δ mutants failed to induce tumor galls when mated with wild-type strains. The discovery of this asymmetric effect on teliospore formation following *trt1* or *ter1* disruption [22,25] is a good reason for further transcriptome analyses to identify the DEGs between these two types of mutants. It will be important to address whether the *ter1* subunit of telomerase can undergo regulation during the dimorphic transition in this fungus. Future work should also determine whether core genes are subject to additional telomerase-independent activities within *U. maydis*.

Data supporting the notion of telomerase deregulation during the dimorphic transition are drawn from [80], wherein slight variations in TERT expression during the shift from sporidia to mycelia development were observed. Those authors also reported the transcriptional repression of *UMAG_03168* in the initial stages of biotrophic development [80]. Intriguingly, *UMAG_03168* encodes the 5′ end of the bicistronic *ter1* transcript [24]. It will be important to establish new cutoff thresholds for determining which genes warrant analysis using RT-qPCR. Additionally, determining suitable housekeeping controls for these genes and devising novel approaches to extract pivotal insights on the regulatory roles of each telomerase subunit in the life cycle of this fungus pose a significant challenge.

## 5. Conclusions

The absence of the RNA component of telomerase in *U. maydis* elicits the upregulation of genes associated with ALT, environmental stress response (ESR), and pathogenesis. The concurrent telomere shortening and loss of TPE may account for the observed ALT and ESR patterns. However, the mechanisms underlying the upregulation of genes located outside subtelomeric regions warrant further investigation. The findings of this study underscore the involvement of TER during the dimorphic transition, potentially suggesting a negative regulatory relationship between Ter1 and the master regulator Rbf1 or other factors that influence its functionality. A comparative analysis of the transcriptomes of *ter1*Δ and *trt1*Δ mutants could shed additional light on potential non-telomeric roles played by the core telomerase subunits, as well as by genes neighboring *ter1*, such as *UMAG_03168*, *UMAG_03169*, and *rbf1*.

## Figures and Tables

**Figure 1 jof-09-00896-f001:**
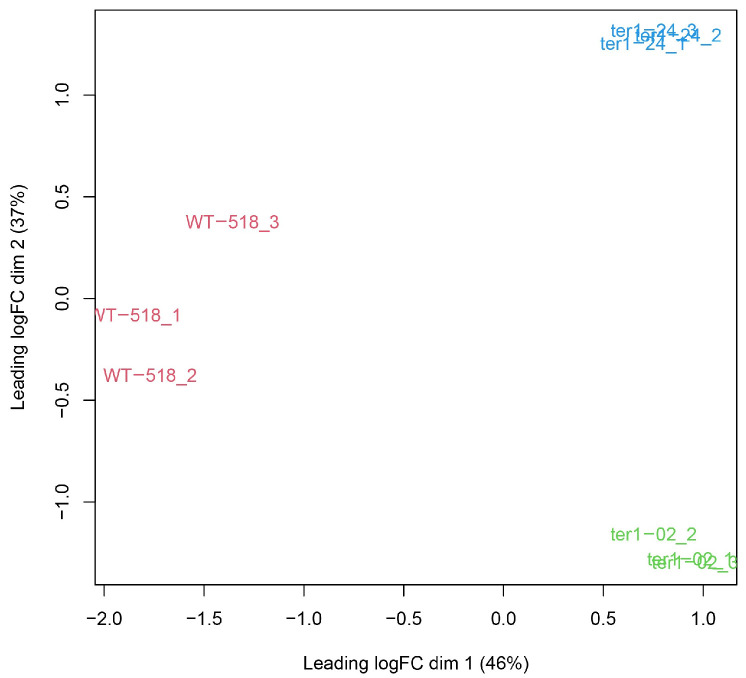
MDS plot. RNA-seq samples are grouped in separate clusters reflecting variations between them.

**Figure 2 jof-09-00896-f002:**
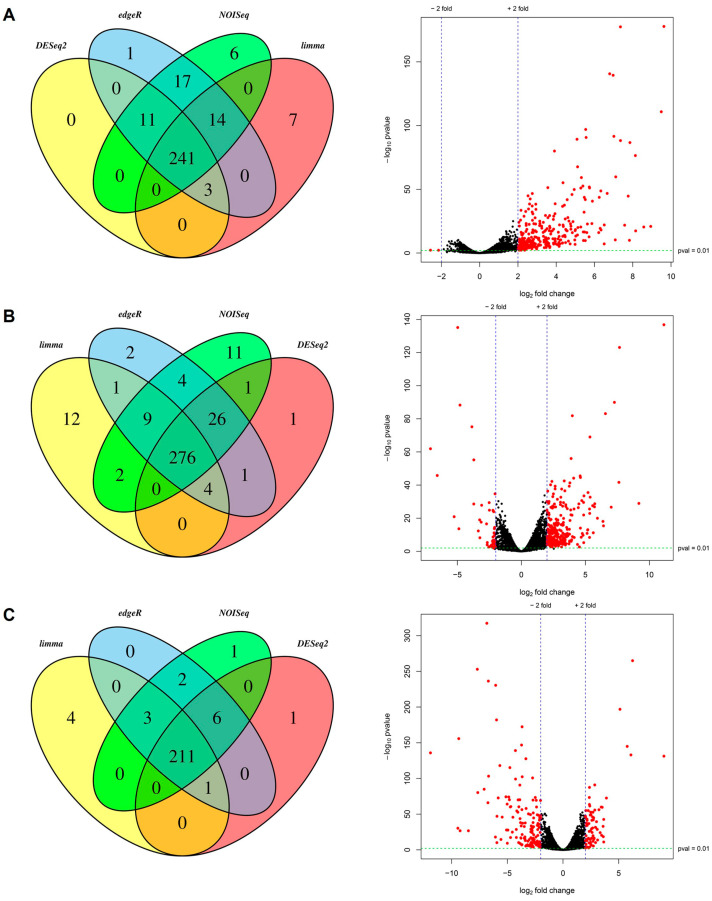
Venn diagrams and distribution of DEGs. The Venn diagrams show the number of DEGs and intersections between each method used. The volcano plots are the distribution of DEGs (red dots) according to the selected cutoff values. (**A**) Differential expression analysis between strain WT 518 and mutant ter1-02. (**B**) Differential expression analysis between strain WT 518 and mutant ter1-24. (**C**) Differential expression analysis between ter1-02 and ter1-24 mutants.

**Figure 3 jof-09-00896-f003:**
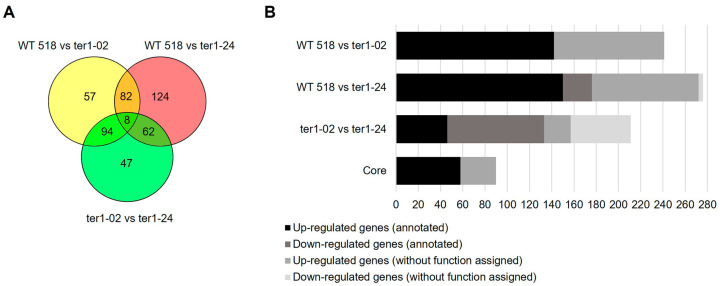
Venn diagram and statistics of the DEGs annotation. (**A**) Representation of the unique and shared DEGs among the analyzed transcriptomes. (**B**) Grouping of DEGs concerning functional annotation assignment.

**Figure 4 jof-09-00896-f004:**
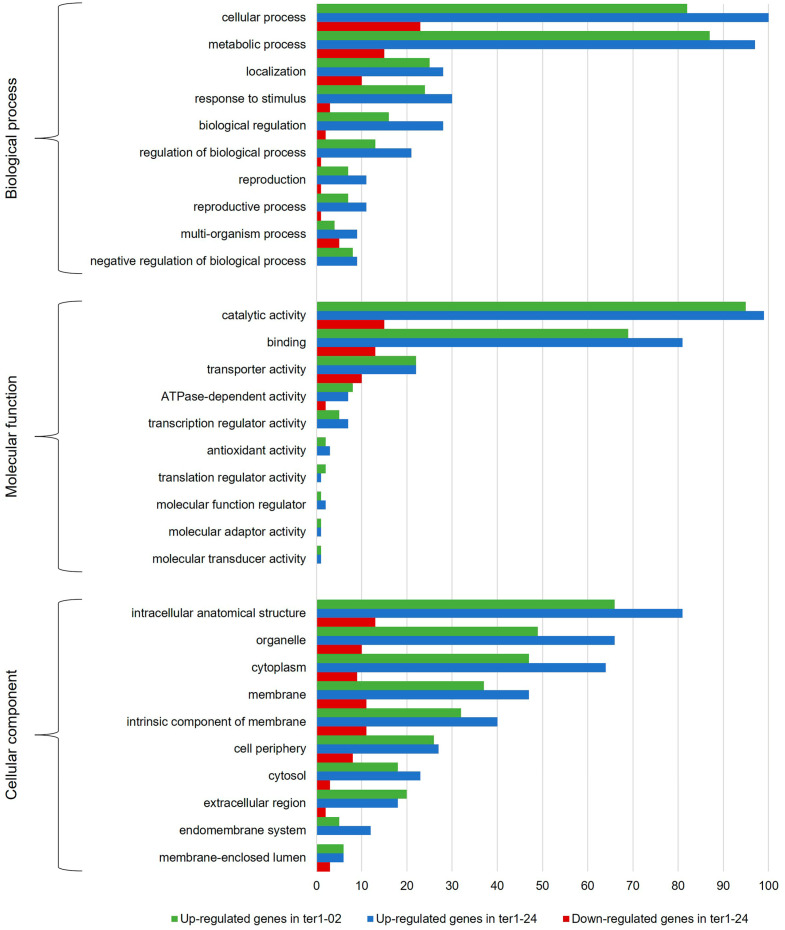
GO classification histogram of the DEGs identified in the ter1 mutants. The graph shows the ten most representative assignments for the categories of biological process (level 2), molecular function (level 2), and cellular component (level 3).

**Figure 5 jof-09-00896-f005:**
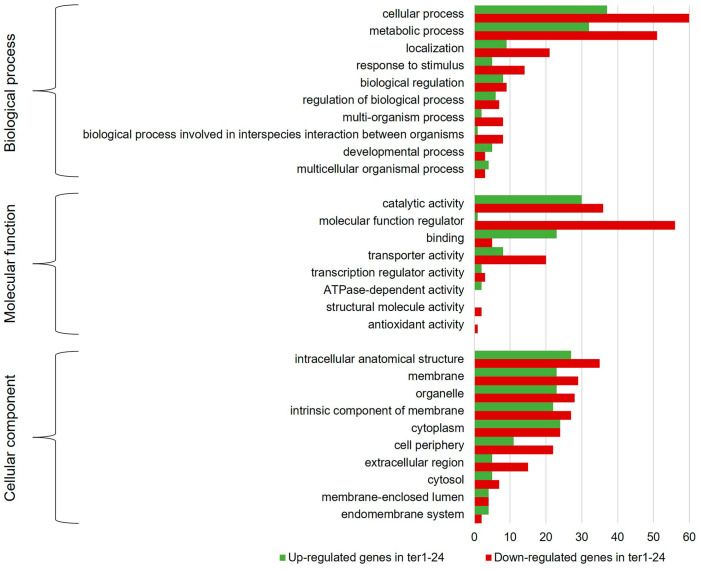
GO classification histogram of the DEGs identified in the differential expression analysis between ter1-02 and ter1-24 mutants. The graph shows the ten most representative assignments for the categories of biological process (level 2), molecular function (level 2), and cellular component (level 3).

**Figure 6 jof-09-00896-f006:**
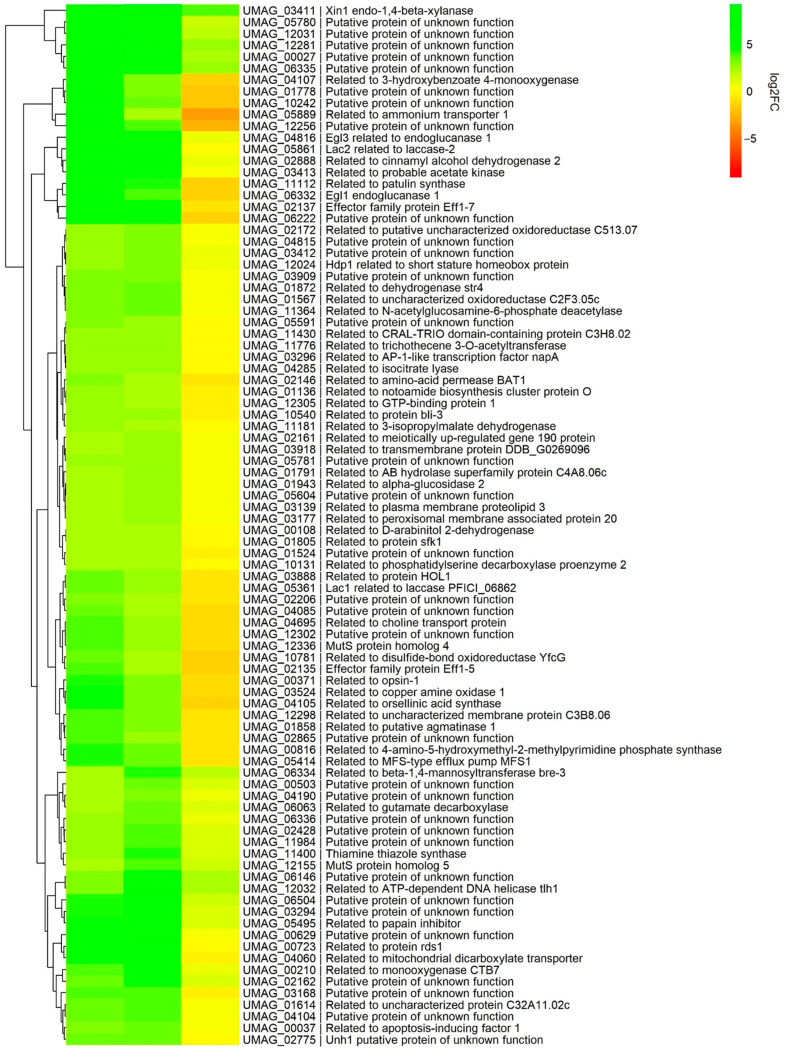
Heat map of the DEGs that comprise the core of the *ter1*::*hph* mutants. The map was constructed with the logFC values obtained from the EdgeR analysis. Comparisons: WT vs. ter1-02 (first column), WT vs. ter1-24 (second column), and ter1-02 vs. ter1-24 (third column).

**Figure 7 jof-09-00896-f007:**
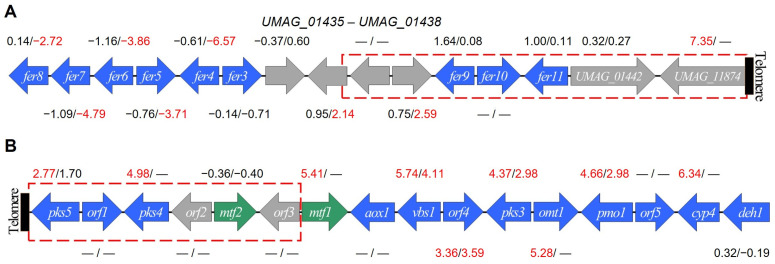
*ter1* disruption causes the deregulation of gene clusters involved in pathogenic development. (**A**) Schematic representation of the iron uptake cluster [61]. (**B**) Schematic representation of the PKS cluster (adapted from [75]). Arrows indicate the direction of gene transcription but not gene sizes. Red rectangles represent the first 20 kb of the chromosomal end. Blue arrows represent genes that are part of the cluster. Gray arrows represent genes not belonging to the cluster’s co-regulated genes. Green arrows represent transcriptional factors. The logFC values obtained from the differential expression analysis with EdgeR are shown. Values on the left correspond to those of strain ter1-02. Values on the right correspond to those of strain ter1-24. DEGs are represented in red.

**Figure 8 jof-09-00896-f008:**
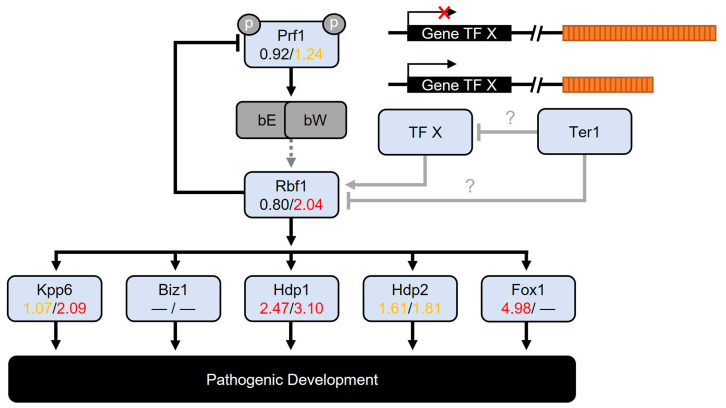
Rbf1 pathway expression is altered in *ter1::hph* mutants. Schematic diagram of the response cascade controlled by the master regulator Rbf1. The logFC values were obtained using the EdgeR analysis, shown below each gene. Values on the left correspond to the ter1-02 strain. Values on the right correspond to the ter1-24 strain. DEGs are represented in red. Genes with an increase in transcriptional expression greater than two-fold are in yellow. ter1-24 shows a significant increase in the expression of Rbf1, independent bW_x_/bE_y_ heterodimer formation (dotted lines); an increase in mainly the expression of Rbf1-target genes is observed in ter1-02. It is tempting to suggest that due to losing TPE as the telomere shortens, an unidentified transcriptional factor (TF X) upregulated its expression. That TF X may have upregulated Rbf1 and promoted the expression of genes downstream of the regulatory cascade. Alternatively, Ter1 could be the negative regulator of Rbf1 or the unidentified TF X (gray arrows with blunted-end heads and question marks).

**Table 1 jof-09-00896-t001:** Telomere-linked RecQ-like helicases are deregulated in *ter1*Δ mutants. The DEGs obtained between the various comparisons are indicated in bold.

Gene Id	Description	WT 518 vs. ter1-02	WT 518 vs. ter1-24	ter1-02 vs. ter1-24
** *UMAG_12032** **	Related to ATP-dependent DNA helicase ^1, 2^	**3.161**	**5.206**	**2.067**
** *UMAG_12076 ^u^* **	Related to ATP-dependent DNA helicase ^1, 2^	**4.939**	0.225	**−4.688**
** *UMAG_06476** **	Related to RecQ helicase ^1, 3^	**4.352**	0.298	**−4.028**
** *UMAG_0647 ** **	Related to RecQ helicase ^1, 3^	**2.924**	0.639	**−2.259**
** *UMAG_11065** **	Related to RecQ helicase ^1, 3^	**6.509**	—	**−9.438**
** *UMAG_04094** **	Related to RecQ helicase ^1, 3^	**8.136**	0.455	**−7.651**
** *UMAG_04308** **	Related to RecQ helicase ^1, 3^	**7.013**	0.308	**−6.679**

^1.^[46], ^2.^[42], ^3.^[43]; * Located in subtelomeric region. ^u^ Unmapped scaffold.

**Table 2 jof-09-00896-t002:** Genes involved in response to environmental stress are deregulated in *ter1*Δ mutants. The DEGs obtained between the various comparisons are indicated in bold.

Gene Id	Description	WT 518 vs. ter1-02	WT 518 vs. ter1-24	ter1-02 vs. ter1-24
** *UMAG_01758* **	Related to multidrug resistance-associated protein 1	**2.756**	0.040	**−2.692**
** *UMAG_01937* **	Related to sphingomyelin phosphodiesterase	1.003	**2.116**	1.136
** *UMAG_01965* **	Related to solute carrier family 40 member 2	1.629	**2.87**	1.263
** *UMAG_02224* **	Related to palmitoyltransferase ZDHHC16	−1.678	1.149	**2.851**
** *UMAG_02753* **	Related to peroxygenase 2	**5.517**	**—**	**−3.186**
** *UMAG_02803* **	Related to beta-1,3-glucan-binding protein	**7.852**	**—**	**−11.899**
** *UMAG_03073* **	Related to glutathione S-transferase 3	1.176	**2.027**	0.881
** *UMAG_03122 ** **	Related to beta-1,3-glucan-binding protein	**2.118**	0.938	−1.161
** *UMAG_03177* **	Related to peroxisomal membrane-associated protein 20	**2.293**	**2.788**	0.519
** *UMAG_03728* **	Related to indoleamine 2,3-dioxygenase 1	−0.804	1.589	**2.421**
** *UMAG_03881* **	Related to 30 kDa heat shock protein	**4.207**	1.426	**−2.755**
** *UMAG_04410* **	Related to MFS siderochrome iron transporter C	−0.794	**−4.970**	**−4.148**
** *UMAG_05600* **	Related to succinate-semialdehyde dehydrogenase [NADP(+)]	**2.335**	1.806	−0.503
** *UMAG_06404* **	Related to peroxiredoxin PRX1, mitochondrial	0.775	−1.938	**−2.688**
** *UMAG_10131* **	Related to phosphatidylserine decarboxylase proenzyme 2	**2.234**	**2.083**	−0.127
** *UMAG_10781* **	Related to disulfide-bond oxidoreductase YfcG	**3.518**	**2.097**	−1.394
** *UMAG_11944* **	Related to glycerol 2-dehydrogenase (NADP(+))	1.495	−0.928	**−2.399**
** *UMAG_12161* **	Related to lipase 5	−0.163	**−2.497**	**−2.305**

* Located in the subtelomeric region.

**Table 5 jof-09-00896-t005:** Transcription factors are deregulated in response to *ter1* mutation. The DEGs obtained between the various comparisons are indicated in bold.

Gene Id	Description	WT 518 vs. ter1-02	WT 518 vs. ter1-24	ter1-02 vs. ter1-24
** *UMAG_01025* **	Related to probable transcriptional regulatory protein STB4	**2.953**	**2.462**	−0.429
** *UMAG_01456* **	Related to regulatory protein CAT8 ^3^	0.758	**2.057**	1.323
** *UMAG_01523* **	Fox1—related to fork head domain transcription factor Slp1 ^1^	**4.985**	**—**	−3.212
** *UMAG_02835* **	Related to conidiophore development regulator abaA ^3^	−0.008	**2.615**	**2.650**
** *UMAG_03296* **	Related to Yap and AP-1-like transcription factor napA	**2.751**	**2.691**	−0.026
** *UMAG_04101 ** **	Mtf1—related to Myb-related protein A ^2^	**5.413**	**—**	−3.400
** *UMAG_04778* **	Related to transcriptional repressor XBP1 ^3^	**—**	**3.538**	1.560
** *UMAG_05721* **	Srb1—related to putative transcription factor sre2 ^3, 4^	−1.494	1.386	**2.904**
** *UMAG_06308* **	Related to transcription factor RFX4 ^3^	0.727	**2.138**	1.432
** *UMAG_12024* **	Hdp1—related to short stature homeobox protein ^5^	**2.476**	**3.108**	0.647
** *UMAG_12304* **	Related to positive regulator of purine utilization	**4.330**	0.912	−3.390
** *UMAG_03172* **	Rbf1—related to zinc finger protein 2 ^6^	0.803	**2.041**	1.249
** *UMAG_02775* **	Unh1 ^7^	**3.315**	**3.281**	−0.011

^1^ [78]. ^2^ [75]. ^3^ [80]. ^4^ [81]. ^5^ [77]. ^6^ [79]. ^7^ [63]. * Located in subtelomeric region.

## Data Availability

Not applicable.

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
