# Peer review of "Global Gene Expression of Post-Senescent Telomerase-Negative ter1Δ Strain of Ustilago maydis"

_jof, 2023, doi:10.3390/jof9090896_

Round 1

Reviewer 1 Report

The work entitles “Global gene expression of post-senescent telomerase negative 2 ter1Δ strain of Ustilago maydis” describes de gene modifications/expression in a mutant strain of U. maydis related to telomerases.  

The work is quite extensive and covers everything related to what the authors are looking for, it is well written although there are some writing problems so I suggest a minor revision.

I can suggest two important aspects:

1) The work is too descriptive, these genes are deregulated, these are activated, these genes are affected, these are not, and the relevance of knowing all this is never mentioned. I consider that the article is too extensive and throughout my review I never understood why the authors wanted to explain this or why it is important to know all this.

2) the discussion is reduced to half a page, since it is such a descriptive work, the reading focuses on what is regulated and what is deregulated (27 pages of the same) while the explanation of this is reduced to a comparison with what was found by [Sanpedro -Luna et al. 2023], so I suggest improving the discussion.

In general, I think it is a work that can be published in its current form.

Author Response

The work entitles “Global gene expression of post-senescent telomerase negative 2 ter1Δ strain of Ustilago maydis” describes de gene modifications/expression in a mutant strain of U. maydis related to telomerases. 

The work is quite extensive and covers everything related to what the authors are looking for, it is well written although there are some writing problems so I suggest a minor revision.

I can suggest two important aspects:

  • The work is too descriptive, these genes are deregulated, these are activated, these genes are affected, these are not, and the relevance of knowing all this is never mentioned. I consider that the article is too extensive and throughout my review I never understood why the authors wanted to explain this or why it is important to know all this.

Dear Sir or madam: Our aim stated in the text. Thank you so much for the suggestion.

  • The discussion is reduced to half a page, since it is such a descriptive work, the reading focuses on what is regulated and what is deregulated (27 pages of the same) while the explanation of this is reduced to a comparison with what was found by [Sanpedro -Luna et al. 2023], so I suggest improving the discussion.

It has been improved; we hope it reaches the standard of you journal. Thanks so much for your suggestion.

Reviewer 2 Report

This is a thorough and authoritative analysis of transcriptional profiling of survivors that arise after deletion of telomerase. The work is carefully performed, appropriately analyzed, nicely presented, and well written.  It is a fine piece of work that will be a resource for investigators interested in telomere/telomerase associated gene expression. It certainly merits publication.

Two minor typos

line 45 -- mistake at theend of sentence--do you mean "senescence sets in"

line 47 -- arise not araise

Author Response

Comments and Suggestions for Authors

This is a thorough and authoritative analysis of transcriptional profiling of survivors that arise after deletion of telomerase. The work is carefully performed, appropriately analyzed, nicely presented, and well written.  It is a fine piece of work that will be a resource for investigators interested in telomere/telomerase associated gene expression. It certainly merits publication.

Two minor typos

line 45 -- mistake at the end of sentence--do you mean "senescence sets in"

The sentence was corrected, thanks you very much for the suggestion. 

line 47 -- arise not araise

The sentence was corrected, thanks you very much for the advice

Reviewer 3 Report

Regarding the manuscript titled "Global gene expression of post-senescent telomerase negative ter1Δ strain of Ustilago maydis" by Sanpedro-Luna et al.

The manuscript presents data that could have been part of a larger manuscript including the characterization of the ter1 deletion mutants but in other organisms, these findings have been separated, that is the deletion of the genes and the subsequent assessment of DEG so this type of presentation does allow comparison.

I have a major concern/criticism that I think should influence all of the presented results and discussion. That is that, as I understand the presentation of data, the authors used two biological reps of the ter 1 U. maydis deletion; however, they do not indicate whether these strains were complemented to show that the deletion actually causes the phenotype and gene expression changes observed. If the deletion strains cannot be complemented it either means that the deletion affected genes other than the target of deletion and as such any data obtained must be considered to result from the deletion of the target gene or the other affected genes or a combination of these. Yet the authors interpret the use of these mutants as if they are clean and only involve the target gene even though they state that the open reading from for ter1 overlaps other genes. I suggest that these mutants, 2 and 24, are in fact distinct that is the insertions in each strain are different, and therefore they are not biological reps of one another and all comparisons using these must be interpreted with this uncertainty.

This suggestion of the differences is supported by observed phenotypic differences in previous studies and the differences in DEGs.

So with the potential lack of biological reps the authors have no independent assessment of DEG and as such all of the data is suggestive.

BUT, if we consider the possibility that the two mutants 2 and 24 actually do target the same gene and that the changes in gene expression are a result of this deletion then we would expect that both deletion strains would exhibit the same impact on gene expression. This is not the case for many of the genes discussed. So the authors must indicate that they are uncertain as to whether the two deletion strains are clean deletions, that the observed alteration in gene expression may be due to a deletion impact on other genes and not ter1, and therefore add caveats to all of their interpretations.  Another possibility might be to acknowledge the uncertainty regarding the deletions strains and state that to be conservative the authors will only consider genes whose expression is similarly altered will be considered as possible responses to ter1 deletion. That means that the authors would alter the results and discussion to only focus on the core genes they identify and acknowledge that they have no support for differential expression of the other genes.

If re-written in this last way it would more accurately reflect the data.

Further, RNA Seq DEG is usually assessed by RT-QPCR to provide an independent confirmation that DEG is actually occurring, the authors did not do this so the interpretation of the results must acknowledge this lack of DEG certainty

I have attached the PDF of the manuscript with comments for some specific edits but given the above concern, the entire results and discussion section will need to be edited substantially.

English is pretty good but the authors must ensure that what they write is what they intend I have indicated some of the areas where I do not think ths is the case in the attached PDF

Author Response

Reviewer 3

Regarding the manuscript titled "Global gene expression of post-senescent telomerase negative ter1Δ strain of Ustilago maydis" by Sanpedro-Luna et al.

The manuscript presents data that could have been part of a larger manuscript including the characterization of the ter1 deletion mutants but in other organisms.

Dear Sir or Madam, this piece of work was initially thought to be part of another article: Sanpedro-Luna, J. A., Jacinto-Vázquez, J. J., Anastacio-Marcelino, E., Posadas-Gutiérrez, C. M., Olmos-Pineda, I., González-Bernal, J. A., . . . Sánchez-Alonso, P. (2023). Telomerase RNA plays a major role in the completion of the life cycle in Ustilago maydis and shares conserved domains with other Ustilaginales. PLoS One, 18(3), e0281251. doi:10.1371/journal.pone.0281251, but reviewers opined that the current manuscript could be published separately during that previous peer review process. The article of Sanpedro Luna et al., 2023 is an original research article carried out with the 518 strain of Ustilago maydis, not with other organisms. Thanks so much for the observation.

these findings have been separated, that is the deletion of the genes and the subsequent assessment of DEG so this type of presentation does allow comparison.

I have a major concern/criticism that I think should influence all of the presented results and discussion. That is that, as I understand the presentation of data, the authors used two biological reps of the ter 1 U. maydis deletion; however, they do not indicate whether these strains were complemented to show that the deletion actually causes the phenotype and gene expression changes observed. If the deletion strains cannot be complemented it either means that the deletion affected genes other than the target of deletion and as such any data obtained must be considered to result from the deletion of the target gene or the other affected genes or a combination of these. Yet the authors interpret the use of these mutants as if they are clean and only involve the target gene even though they state that the open reading from for ter1 overlaps other genes.

Dear Sir or Madam, ter1Δ complementation is technically unfeasible because ter1Δ survivors are very frail; other challenges rely on the cell wall composition, which changes in the mutants, possibly because of the deregulation of genes involved in its synthesis. That fact practically impedes the obtention of living protoplasts to achieve gene transformation because cells die after prolonged incubation in CW-lysing enzymes. Cold temperatures to transform cells could also be lethal, and the replication times are so large that we would observe transformants (if any) after a week.

Indeed, transcription of ter1 initiates 5’ upstream from the intergenic region, where most of the gene is positioned, between UMAG_03168 and UMAG_03169; previous analysis indicates the existence of several isoforms that originates 5’ from UMAG_03168 or overlap with it, but the disruption in the intergenic gene of ter1 did not interrupt the transcription of main isoforms of the putative gene UMAG_03168; i.e., isoform A (not spliced isoform: logFC = 0.47 in ter1-02, and lof FC =1.9 in ter1-24) and isoform B (spliced transcript: logFC = 3.9 in ter1-02, and logFC = 3.6 in ter1-24); in fact, we are studying the contribution of the hypothetic protein-encoding gene to a phenotype. The UMAG_03169 gene, which encodes a putative ornithine-oxo-acid transaminase, also had upregulation (logFC = 1.74 in ter1-02 and logFC = 3.0 in ter1-24) in the ter1-disrupted mutants. Some transcript isoforms of the non-coding ter1 gene (which in a strict sense does not have an ORF) also were found; no polar effect by ter1-disruption was caused on surrounding genes; thanks for the advice.

I suggest that these mutants, 2 and 24, are in fact distinct that is the insertions in each strain are different, and therefore they are not biological reps of one another and all comparisons using these must be interpreted with this uncertainty.

AND So with the potential lack of biological reps the authors have no independent assessment of DEG and as such all of the data is suggestive.

Dear Sir or Madam, respectfully, we differ in opinion. Please see the materials and methods of the article of Sanpedro-Luna et al., 2023; the reviewer will find here the procedure to evidence that both mutant strains are disrupted in the same site could be found. Thanks for the commentary.

This suggestion of the differences is supported by observed phenotypic differences in previous studies and the differences in DEGs.

Please be aware that telomerase-negative mutants are lethal; after most of the cells die, some survivors occasionally arise, which maintains telomere function by recombination-based mechanisms. That machinery is also influenced by epigenetic changes in the chromatin, as those that control the dimorphic transition from yeast-like cells to mycelium. Epigenetic mechanisms seem to underlie the type of survivors (type I or type II) the mutant chooses to survive. As you will see in the improved discussion, we hypothesized that the reason for the differences in the phenotype in the mutant strains would be derived from epigenetic disturbances.

We also mentioned that a point mutation is in the 3’ outside of the TER gene on the same intergenic region (not affecting other genes). However, we are taking the research steps to confirm this could cause the variation. Thanks for the revision.

So with the potential lack of biological reps the authors have no independent assessment of DEG and as such all of the data is suggestive.

BUT, if we consider the possibility that the two mutants 2 and 24 actually do target the same gene and that the changes in gene expression are a result of this deletion then we would expect that both deletion strains would exhibit the same impact on gene expression.

This is not the case for many of the genes discussed. So the authors must indicate that they are uncertain as to whether the two deletion strains are clean deletions, that the observed alteration in gene expression may be due to a deletion impact on other genes and not ter1, and therefore add caveats to all of their interpretations. 

Dear Sir or Madam. We made sure by PCR amplification of both sides of the disrupted template domain that the reps were in the same site; we waited for its stabilization after several passages, we confirmed by Southern blot the overlengthening of telomeric sequences and checked the karyotype by PFGE (materials would be sent on request).

We are reporting the effect of the ter1-mutation on the expression of genes involved in surviving after the lack of telomere maintenance. In response to this condition, several genes involved, such as the general stress response, genotoxic and abiotic stress response, cell cycle regulation, and others, operate to give stability to eukaryote genomes in those circumstances.

As we stated, this fungus experience differentiation, and the results open the possibility of exploring the regulation of TER in lower eukaryotes that experience developmental transitions.

Further, RNA Seq DEG is usually assessed by RT-QPCR to provide an independent confirmation that DEG is actually occurring, the authors did not do this so the interpretation of the results must acknowledge this lack of DEG certainty.

Dear Sir or Madam. Our results matched, in general, with the obtained by several working groups with fungal survivors of telomerase-RNA mutants. Despite differences, both survivors exhibited core DEGs; please consider that although this is not the case, the deregulation of TDR genes is still not leveled during the survivor transition, reaching a maximum of TDR genes when telomere length is shortest. However, in the future, we will be working at that point. Thanks for the suggestions.

I have attached the PDF of the manuscript with comments for some specific edits but given the above concern, the entire results and discussion section will need to be edited substantially.

All the points have been aboarded, thanks for your aid.

Round 2

Reviewer 3 Report

The authors have appropriately responded to the comments made regarding the original manuscript. The authors should consider that some of the edits they have made introduced very long sentences and that reading these long sentences can sometimes obscure the message that the authors are attempting to convey. So the authors may want o edit long sentences to further improve the clarity of the edits made. The more full discussion of the potential implications of the results has clarified the manuscript.

Long sentences need to be edited to shorten and clarify the messages but in general the improvements are ok.

Author Response

Dear Sir or Madam:

Thank you for your new review. The manuscript has been reviewed to shorten long sentences; also, the changes therein are highlighted as required by the editor; other changes to clarify the meaning also were done.

Thank you for your suggestions